# Prediction of Histological Grade of Oral Squamous Cell Carcinoma Using Machine Learning Models Applied to ^18^F-FDG-PET Radiomics

**DOI:** 10.3390/biomedicines12071411

**Published:** 2024-06-25

**Authors:** Yutaka Nikkuni, Hideyoshi Nishiyama, Takafumi Hayashi

**Affiliations:** Division of Oral and Maxillofacial Radiology, Graduate School of Medical and Dental Sciences, Niigata University, Niigata 951-8510, Japan; nisiyama@dent.niigata-u.ac.jp (H.N.); hayashi@dent.niigata-u.ac.jp (T.H.)

**Keywords:** radiomics, machine learning model, ^18^F-FDG-PET, oral squamous cell carcinoma, histological grade

## Abstract

The histological grade of oral squamous cell carcinoma affects the prognosis. In the present study, we performed a radiomics analysis to extract features from ^18^F-FDG PET image data, created machine learning models from the features, and verified the accuracy of the prediction of the histological grade of oral squamous cell carcinoma. The subjects were 191 patients in whom an ^18^F-FDG-PET examination was performed preoperatively and a histopathological grade was confirmed after surgery, and their tumor sizes were sufficient for a radiomics analysis. These patients were split in a 70%/30% ratio for use as training data and testing data, respectively. We extracted 2993 radiomics features from the PET images of each patient. Logistic Regression (LR), Support Vector Machine (SVM), Random Forest (RF), Naïve Bayes (NB), and K-Nearest Neighbor (KNN) machine learning models were created. The areas under the curve obtained from receiver operating characteristic curves for the prediction of the histological grade of oral squamous cell carcinoma were 0.72, 0.71, 0.84, 0.74, and 0.73 for LR, SVM, RF, NB, and KNN, respectively. We confirmed that a PET radiomics analysis is useful for the preoperative prediction of the histological grade of oral squamous cell carcinoma.

## 1. Introduction

The World Cancer Research Fund International reported more than 377,700 oral cancer cases worldwide in 2020, making it the 16th most common cancer among all cancers, the 11th most common in men, and the 18th most common in women [1]. The age-standardized rate of oral cancer incidence was 8.0 per 100,000 people worldwide, and the number of deaths from oral cancer was 3.9 per 100,000 people (age-standardized rate). Surgical resection is the first treatment strategy for oral cancer, but in recent years, intra-arterial chemotherapy has attracted attention [2]. Many researchers have reported that the histological grade affects the prognosis of oral squamous cell carcinoma after treatment [3,4,5]. For this reason, the histological grade of the tumor is determined by a biopsy prior to treatment, but this is an invasive procedure and carries the risk of tumor expansion. If a non-invasive accurate preoperative diagnosis of the histological grade of oral squamous cell carcinoma is possible, it will greatly contribute to the treatment of oral cancer. Imaging examinations are essential for preoperative stage determination and for postoperative follow-up of oral cancer. Among the various imaging examination methods used for this purpose, which include computed tomography (CT), magnetic resonance imaging (MRI), and ultrasonography (US), the importance of positron emission tomography (PET), a type of nuclear medicine examination, has recently increased [6]. ^18^F-FDG-PET is an imaging method that uses the metabolic trapping of 6-fluorodeoxyglucose (FDG) to accumulate the radioactive isotope ^18^F within cells, and this accumulation is then used to map the three-dimensional (3D) distribution of metabolism in tissues. The standardized uptake value (SUV), which is a semi-quantitative value representing local FDG accumulation, is often used to evaluate FDG-PET examinations. PET examinations currently play a very important role in determining the stage of oral cancer [7], with several researchers reporting that FDG-PET is useful for predicting the prognosis of oral cancer. For this purpose, the maximum value (SUVmax) or mean value (SUVmean) of a region of interest, the metabolic tumor volume (MTV), and total lesion glycolysis (TLG) are typically used [8,9,10]. 

With the development of different imaging modalities, the amount of digital information obtained during workup has increased dramatically in recent years, and it has become possible to three-dimensionally display a variety of information within the body with fine spatial resolution. To effectively use this large amount of imaging data, the radiomics method has been attracting attention. Radiomics is a method that comprehensively extracts a large number of quantitative features from images, making it possible to evaluate image information in more detail than before [11,12]. One method for evaluating such large amounts of data is to use machine learning models. Machine learning models, which are used in various fields, are prediction algorithms that use multiple features related to the prediction target to make predictions. They are typically created using training data, with their accuracy then being verified using separate testing data. Much research has applied machine learning models to the medical field [13]. The large number of features extracted by radiomics methods can be handled more effectively by machine learning models than by manual evaluation. Examples of radiomics research using machine learning models include the prediction of recurrence after the treatment of malignant tumors and the prediction of subsequent cervical lymph node metastasis [14,15]. In terms of imaging modality, radiomics and machine learning models have been applied to a variety of image types, including CT and MRI, and the usefulness of radiomics analysis using machine learning models was reported for the evaluation of malignant tumors on PET images [16]. Radiomics features obtained from PET images may provide more useful information than conventional methods for the preoperative diagnosis of oral cancer. On the basis of the above, we hypothesized that radiomics features extracted from PET images can predict the histological grade of oral squamous cell carcinoma. We verified this hypothesis by evaluating the accuracy of different machine learning models using radiomics features. Through this verification, the purpose of this study was to examine whether this technique, which has the advantage of being non-invasive and can be performed preoperatively, can replace the role of a biopsy.

## 2. Materials and Methods

### 2.1. Ethical Approval

This study was approved as epidemiological research in humans after ethical review by the ethics committee of Niigata University (approval No. 2022-0327). The study procedures were in accordance with the Helsinki Declaration. Verbal informed consent was obtained from all patients. 

### 2.2. Subjects

The subjects included in this retrospective study met the following criteria: (1) a clinical diagnosis of oral cancer at Niigata University Medical and Dental Hospital between July 2016 and July 2022; (2) ^18^F-FDG PET/CT examination before treatment; (3) resection selected as a treatment strategy; and (4) a primary tumor histopathologically diagnosed as squamous cell carcinoma after surgery, with the histological grade determined. Cases for which PET examination was not performed before treatment, for which the histological grade could not be determined histopathologically, and for which it was not possible to extract radiomics features from PET images were excluded.

### 2.3. Image Acquisition

^18^F-FDG PET/CT was performed using a biograph mCT Flow 20 (SIEMENS Healthineers, Erlangen, Germany) scanner for preoperative imaging diagnosis. The imaging range was from the skull base to the middle of the thigh, and the spatial resolution was 4.07 × 4.07 × 2.00 mm. 

### 2.4. Radiomics Analysis

The radiomics analysis was performed using the following steps: (1) segmentation of lesions; (2) extraction of radiomics features from each segmented lesion; (3) identification of the radiomics features that are useful for predicting the histological grade of oral squamous cell carcinoma; (4) radiomics feature selection for creating the machine learning prediction models; and (5) evaluation of oral squamous cell carcinoma histological grade prediction accuracy using machine learning models.

#### 2.4.1. Segmentation of Lesions

To obtain radiomics features from image data, we segmented regions of interest. To obtain radiomics features from 3D images, the images were first transformed to an isotropic voxel size of 1 mm to prevent bias and distortion from affecting the features obtained from a specific cross section. Segmentation was performed on all slices containing the target using 3D slicer version 5.3.0 software (an open-source software platform for medical image data). Figure 1 shows an example of segmentation of a lesion. As a preprocessing step for the segmentation, relevant image portions were first extracted using an SUV threshold of 2.5 (a value used by many researchers to segment neoplastic lesions), and then one oral and maxillofacial radiology specialist with 19 years of experience (Y. N.) manually extracted the region corresponding to the lesion on two separate days with reference to clinical information.

#### 2.4.2. Extraction of Radiomics Features from Each Segmented Lesion

Radiomics features were obtained from the segmented region using the open-source Python package PyRadiomics version 3.0.1. Radiomics features were classified into three types: shape features, first-order features, and texture features. Shape features were digitized morphological features of a 3D image, and 14 of these were defined. There were 18 first-order features, including the average value, minimum value, maximum value, kurtosis, and skewness, which were obtained from the histogram of the pixel value distribution of the region of interest. The radiomic texture features were numerical values allowing for a quantitative evaluation of the texture pattern of a two-dimensional image. In reality, for each pixel (voxel) of an image, a matrix was created based on the difference and continuity of the values of adjacent pixels (voxel), and feature quantities were extracted from this matrix. The metabolic distributions of the tumors depicted in the PET images involved 3D image information, and the texture features represented the relationship between one voxel and 28 three-dimensionally adjacent voxels. The texture features adopted in this study were extracted from each of the following five matrices: (1) Gray Level Co-occurrence Matrix (GLCM); (2) Gray Level Dependence Matrix (GLDM); (3) Gray Level Run Length Matrix (GLRLM); (4) Gray Level Size Zone Matrix (GLSZM); and (5) Neighboring Gray Tone Difference Matrix (NGTDM). (1) The GLCM is a matrix of numerical differences between two adjacent voxels, and 24 features were extracted from this matrix. (2) The GLDM is a matrix of the density/range of voxels included within a certain density range surrounding a voxel, and 14 features were extracted from this matrix. (3) The GLRLM is a matrix focusing on voxels that have the same pixel value and are continuous in one direction, and 16 features were extracted from this matrix. (4) The GLSZM is a matrix focusing on the area size and pixel value of consecutive voxels with the same pixel value, and 16 features were extracted from this matrix. (5) The NGTDM is related to the sum of absolute differences between neighboring voxels, and five features were extracted from this matrix. For the first-order and texture features, each image was wavelet transformed with the spatial frequency composed of high-frequency components in the X, Y, and Z directions (labeled WL-HHH) and with low-frequency components in all three directions (labeled WL-LLL). PET images have the potential to reflect the histological features of a lesion in the form of both very small and large changes in signal intensity, which may be captured by texture features. In order to obtain texture features from a PET image, it is necessary to divide (binning) the SUV distribution on the PET image at a certain boundary value, and this interval is called “bins”. To capture small changes in the SUV as a texture feature, the bins should be made small, and to capture large changes in the SUV as a texture feature, the bins should be large. Therefore, texture features were obtained for each bin of 0.01, 0.02, 0.03, 0.05, 0.1, 0.2, 0.3, 0.5, 1, 2, 3, 5, and 10. 

A total of 2993 radiomics features were finally obtained, including 14 shape features, 54 first-order features, and 2925 texture features. All acquired radiomics features were normalized for subsequent processing by z-score normalization, i.e., they were transformed to a mean of 0 and a standard deviation of 1, to eliminate scale differences in each feature. Next, to evaluate the reproducibility of the z-scored features, we calculated the intraclass correlation coefficient (ICC) between the values obtained in the first and second segmentations. Only features with a value of 0.75 or higher, indicating a high correlation, were adopted.

#### 2.4.3. Verification of the Radiomics Features Useful for Predicting the Histological Grade of Oral Squamous Cell Carcinoma

The measured features were divided into two groups based on histological grade, namely a well-differentiated (squamous cell carcinoma) group and a moderately/poorly differentiated group, and Welch’s *t* test was used to identify features with a significant difference of *p* < 0.05 between the groups. 

#### 2.4.4. Radiomics Feature Selection for Creating Machine Learning Prediction Models

To eliminate redundancy in the adopted features and to ensure that each feature was independent from other features, we calculated the Spearman correlation coefficients between all pairs of features that showed a significant difference in Welch’s *t* test, and if any two features had a value of 0.9 or higher, only one of the features was adopted. To create machine learning models with high generalization ability for the adopted features, we used the Least Absolute Shrinkage and Selection Operator (LASSO) regression model to reduce the number of features. This is an L1 regularization-based method that penalizes the L1 norm of the feature weighting factors. LASSO regression reduces features by shrinking all regression coefficients towards zero and setting the coefficients of uncorrelated features to zero. After LASSO feature screening, the final features were selected to build Logistic Regression (LR), Support Vector Machine (SVM), Random Forest (RF), Naïve Bayes (NB), and K-Nearest Neighbor (KNN) machine learning models. LR is a model that uses multiple features to predict one of two or more discrete classes. This model is a regression analysis that takes multiple features as input and calculates a function that outputs the probability that the prediction will be true. SVM is a machine learning model that uses margin maximization and the kernel method to create a “classifier” that determines a boundary line or hyperplane dividing two classes of training data. This model calculates separation using a hyperplane that maximizes the distance (margin) from the sample closest to the boundary. The kernel function is an algorithm used to increase dimensionality and separation by adding nonlinear features to data representation. RF is a machine learning model that creates many decision trees and predicts the classification of unknown data according to the majority vote of the decision trees. NB is a machine learning model that calculates the probabilities of all estimates of given data and outputs the one with the highest probability as the estimation result. It is assumed that the data features are independent and uncorrelated with each other, and that each feature independently affects the estimation results. KNN is a method of dividing data into groups by inferring to which group the target data belongs using a majority vote of the surrounding data.

#### 2.4.5. An Evaluation of the Accuracy of the Machine Learning Models’ Predictions of Oral Squamous Cell Carcinoma Histological Grade

The patients were randomly divided into portions of 70% and 30%, and the 70% group was used as a training dataset to build the machine learning models. The remaining 30% group served as the testing dataset for the built models. The diagnostic performance of each machine learning model was determined by evaluating the receiver operating characteristics curve (ROC), area under the curve (AUC), accuracy, sensitivity, specificity, positive predictive value (PPV), and negative predictive value (NPV).

## 3. Results

Figure 2 shows the flow chart of subject inclusion in this research. Overall, 191 patients were included, with these having a mean age of 68.9 (±12.5) years and consisting of 113 men with a mean age of 68.0 (±11.7) years and 78 women with a mean age of 70.3 (±13.6) years. Of the 191 patients, 146 were assigned to the well-differentiated group, and 45 were assigned to the moderately/poorly differentiated group. The primary tumor sites in all patients were the tongue (n = 97), floor of the mouth (n = 10), maxillary gingiva (n = 22), mandibular gingiva (n = 38), buccal mucosa (n = 22), palate (n = 1), and lip (n = 1). The T classifications of the primary tumors were T1 for 69 patients, T2 for 65, T3 for 12, T4a for 42, and T4b for 3. In the 191 included patients, the average radioactivity at the time of tracer administration in the PET examination was 259.5 MBq, the average administered blood glucose level was 107.4 mg/dl, and the average waiting time before commencing image acquisition was 54.7 min. Eleven patients were excluded, including six in which the tumor was not visualized when the SUV threshold was set to 2.5, and five in which the segmented area was too small to allow for the extraction of radiomics features. Table 1 lists the characteristics of the patients included in this study. 

Among the 2993 extracted radiomics features, 39 showed a significant difference between the well-differentiated group and the moderately/poorly differentiated group in the Welch’s *t* tests, while the Spearman correlation coefficients were less than 0.9 for all feature pairs. Of the 39 features showing significant differences, 1 was a first-order feature, and 38 were texture features; no morphological features showed a significant difference. Among the texture features, 12 were from the GLCM, 6 were from the GLDM, 8 were from the GLRLM, 9 were from the GLSZM, and 3 were from the NGTDM; 2 of these texture features were extracted from the original images, 32 were from the wavelet-HHH (high frequency) images, and 5 were from the wavelet-LLL (low frequency) images. In addition, most of the features obtained were from bins 0.5, 0.1, and 0.2, with significant differences between the smallest bin of 0.01 and the largest bin of 10. These features are listed in Table 2. The LASSO regression resulted in nine features being selected for the construction of the machine learning models: bins0.01 WL_HHH Histogram Mean, bins0.5 WL_HHH GLCM Idmn, bins0.05 WL_HHH GLCM Cluster Shade, bins0.1 WL_LLL GLDM Dependence Non-Uniformity Normalized, bins5 WL_HHH GLRLM Short Run High Gray Level Emphasis, bins0.5 WL_HHH GLRLM Short Run High Gray Level Emphasis, bins0.05 WL_HHH GLSZM Small Area Low Gray Level Emphasis, bins0.5 WL_HHH GLRLM Short Run High Gray Level Emphasis, and bins0.5 WL_HHH NGTDM Complexity. Table 3 shows the selected features and their respective coefficients. 

Table 4 shows the performance of each machine learning model. For the training cohort, the AUC, accuracy, sensitivity, specificity, PPV, and NPV were 0.72, 81%, 12%, 98%, 60%, and 82%, respectively, for the LR model; 0.73, 76%, 13%, 100%, 100%, and 78% for the SVM model; 0.98, 94%, 77%, 99%, 96%, and 94% for the RF model; 0.78, 74%, 51%, 82%, 50%, and 83% for the NB model; and 0.72, 78%, 0%, 100%, 0%, and 78% for the KNN model. For the testing cohort, the AUC, accuracy, sensitivity, specificity, PPV, and NPV were 0.72, 67%, 11%, 100%, 100%, and 100%, respectively, for the LR model; 0.71, 77%, 15%, 98%, 67%, and 80% for the SVM model; 0.84, 79%, 53%, 88%, 62%, and 84% for the RF model; 0.74, 72%, 70%, 72%, 35%, and 92% for the NB model; and 0.73, 74%, 13%, 100%, 100%, and 76% for the KNN model. Figure 3 and Figure 4 show the respective ROC curves of each machine learning model on the training and testing data.

## 4. Discussion

No shape features showed a significant difference between the two groups. The volume of the region of interest above a certain uptake threshold in the PET images is known as the metabolic tumor volume (MTV), and it has been reported to be useful for evaluating tumor aggressiveness [9]. Among the radiomics features extracted in this work, the shape feature “volume” corresponds to the MTV, but we did not find that it was useful for differentiating the histological grade. This result suggests that the histological grade of oral squamous cell carcinoma cannot be determined by only morphological features defined by a threshold of 2.5 when the degree and distribution of internal metabolism are excluded. In addition, this indicates that it is difficult to predict the histological grade of a tumor from its size and extent on FDG-PET images. Of the thirty-nine features that showed significant differences, all but one of them (38) were texture features, indicating that differences in the distribution of metabolic areas inside the tumor reflected the tissue type. The texture features with significant differences included a variety of bins from 0.01 to 10, suggesting that the distributions of both small and large differences in SUV reflected the histological grade. Among the first-order features, there was a significant difference between the two groups in the “mean” feature. An attempt to predict the histological grade of oral cancer from PET images was previously performed at our institution [17]. This previous work involved a histogram analysis, and a statistically significant difference was observed in the kurtosis values between the well-differentiated group and the moderately/poorly differentiated group. In this study, we did not obtain results similar to those in previous studies; that is, we did not find significant differences between the two groups in the features that indicated the properties of the histogram. In this study, the PET images were subjected to isotropic voxel processing for the accurate extraction of texture features, and this may have been at least partly responsible for the differences in results. The nine features extracted for use in the machine learning models included seven features obtained from WL-HHH images and two features obtained from WL-LLL images. The WL-HHH images captured regions with high spatial frequency, that is, regions with large changes in the SUV, and we think that these corresponded to the peripheral region of the tumor. In contrast, the WL-LLL images captured regions with low spatial frequency, that is, regions with few changes in the SUV, which seemed to correspond to the center of the tumor. Considering the relationships between the two types of features separated by this wavelet and the actual histopathological features, the area of the tumor margin indicated by WL-HHH would have included the tumor tissue and normal tissue. The three texture features, GLRLM Short Run High Gray Level Emphasis, GLCM Idmn, and NGTDM Complexity, created with bin 0.5 in the area indicated by WL-HHH, were features characterized by common SUV differences. GLRLM Short Run High Gray Level Emphasis reflects a high SUV in a short region, GLCM Idmn reflects local uniformity, and NGTDM Complexity reflects non-uniformity and rapid changes. These feature values indicate, first, that there was a region with a uniform SUV in a portion corresponding to the lesion margin. Additionally, a rapid change in the SUV was observed in this region, which suggests that although there were multiple regions with uniform SUVs, their number was small. In other words, this suggests that one of the uniform SUV regions was normal tissue, and the other was tumor tissue. As described above, a tumor region with a high SUV was situated between the texture area that appeared to reflect a tumor region and the texture that appeared to reflect normal tissue, and this was reflected by the GLRLM Short Run High Gray Level Emphasis feature. This GLRLM Short Run High Gray Level Emphasis feature clearly indicated the highly differentiated group, even when looking at the large SUV difference of bin 5, and the boundary between normal tissue and tumor tissue was very clear at the lesion margin shown by WL-HHH. Furthermore, among the texture features selected from the lesion area indicated by WL-HHH, the GLCM Cluster Shade and GLSZM Small Area Low Gray Level Emphasis features reflected a small change in bins of 0.05. The feature GLCM cluster Shade indicated that the distribution of SUVs within the region was not uniform but uneven, and GLSZM Small Area Low Gray Level Emphasis indicated that many small regions with low SUVs were present. In other words, in well-differentiated squamous cell carcinoma, a small area with a low SUV is characteristically recognized as an element forming a texture in the lesion margin area. The basis for classifying oral squamous cell carcinoma into well-differentiated, moderately differentiated, and poorly differentiated tissues depends on the degree of keratinization. Well-differentiated squamous cell carcinoma is highly keratinized and forms cancer pearls. Because keratinized materials such as cancer pearls do not participate in metabolism, it is reasonable to expect that tumor regions containing them will have lower metabolism than regions that do not contain them. In other words, the SUV of the well-differentiated group might be lower than that of the moderately/poorly differentiated group. However, while there was no significant difference in the mean or maximum values of the first-order features between the two groups, the GLSZM Small Area Low Gray Level Emphasis feature was significantly higher in the highly differentiated group, that is, there were many small areas with a low SUV in the tumor areas. We think that the GLSZM Small Area Low Gray Level Emphasis and GLCM cluster Shade features composed of 0.05 bins indicated the presence of keratinized material within the lesion. The region at the center of the tumor indicated by WL-LLL contained tumor cells and their stroma. In this region, the GLDM Dependence Non-Uniformity Normalized feature, which consists of 0.1 bins, was selected as a significant feature value in the moderately/poorly differentiated group, and the GLSZM Small Area High Gray Level Emphasis feature, which consists of 10 bins, was selected as a significant feature value in the well-differentiated group. The GLDM Dependence Non-Uniformity Normalized feature indicates SUV heterogeneity within a certain range of adjacent regions, and we think that it is a texture feature reflecting the internal heterogeneity of moderately/poorly differentiated squamous cell carcinoma. The texture feature indicated by the GLSZM Small Area High Gray Level Emphasis feature seemed to reflect the area of high SUV near the center of the area where FDG accumulated in the tumor area. This region appeared to represent wide areas with low values in tumors with heterogeneous internal metabolism and small areas with high values in tumors with homogeneous internal metabolism. Radiomics analyses conducted using other modalities or on malignant tumors at other sites have suggested a relationship between tumor aggressiveness and internal heterogeneity indicated by texture features [18,19,20,21,22,23]. Many machine learning models showed results with low sensitivity and high specificity, and most models had a low PPV and a high NPV. These results indicate that most of the machine learning models were good at predicting the well-differentiated group but had difficulty predicting the moderately/poorly differentiated group. This suggests that many of the metabolic characteristics of the moderately/poorly differentiated group overlapped with those of the well-differentiated group and that there were few unique metabolic features in the moderately/poorly differentiated group. To correct this tendency and improve the accuracy of the machine learning models, it was necessary to identify and select radiomics features that reflected the histopathological characteristics of moderately/poorly differentiated squamous cell carcinoma. On the basis of past reports of PET radiomics analysis for the histological diagnosis of other malignant tumors, it is highly likely that the feature values reflected the heterogeneity of SUVs [24]. The results of this study suggest that the tumor margin area shown by WL-HHH contained many texture features that could differentiate the histological grade, and that when distinguishing the histological grade of squamous cell carcinoma, more accurate predictions could be made by extracting a portion of the tumor margin rather than the entire tumor. A report on MRI radiomics analysis by Fei et al. stated that segmentation that included a 10 mm circumference around the tumor enabled a more accurate diagnosis of its aggressiveness [25]. In addition, Konishi et al. reported that the diagnostic accuracy increased when the area around the metastatic lymph nodes was included in the US radiomics analysis [26]. Thus, when evaluating the properties of a tumor, it is necessary to consider segmentation methods, such as whether or not to segment only the tumor margin or include the tumor margin within the segmentation area. 

This may not only improve the diagnostic accuracy of the histological grade, but also contribute to the realization of personalized medicine, which selects the optimal treatment for each case based on its relationship with prognosis. Combining the prediction of the histological grade using radiomics and biomarkers and genetic research enables a multi-omics analysis to be carried out, which may improve the accuracy of the histological grade diagnosis.

The present study had three main limitations. First, the number of eligible cases was only 191, and the included cases were collected from a single facility. To obtain machine learning models with higher accuracy and robustness, it is necessary to collect more cases and cases from other institutions. The second limitation was the low spatial resolution of the PET images used in this study. Research involving analyses of the texture features of neoplastic lesions has been conducted using various modalities, such as CT, MRI, US, and PET, but among them, PET images have the lowest spatial resolution. Therefore, there is a possibility that the PET images did not accurately reflect the texture features of the tumor. Additionally, studies have reported that the low spatial resolution of PET has a negative effect on understanding intratumoral heterogeneity [27,28]. If PET equipment with higher spatial resolution becomes available in the future, it may be possible to identify texture features that reveal the histological grade of oral squamous cell carcinoma more clearly than those identified in this study. The third limitation was that the method of predicting the tissue type by building an ML model from PET radiomics features is incomparable in terms of cost compared to predicting the histological grade through a biopsy. However, since the present method was used to non-invasively predict the histological grade and has proven to be highly accurate, it is desirable to proceed with studies to determine whether the cost can be saved and reduced in the future.

## 5. Conclusions

We confirmed that machine learning models composed of selected radiomics features derived from ^18^F-FDG PET images were useful for the preoperative histological diagnosis of oral squamous cell carcinoma. Using an ML model with preoperative PET radiomics features can be a non-invasive way to diagnose the histological grade compared to using conventional biopsy alone. The more we can achieve accurate histological diagnoses through this combined diagnostic method, the more accurate the prediction of prognosis may become than ever before. Furthermore, by linking the more accurate histological diagnosis with the clinical data of histological findings, such as marked nuclear and cellular pleomorphism, nuclear hyperchromasia, and mitotic figures (including atypical forms), and small islands or individual cells observed at the invasive front indicated by the WHO classification, it is possible to select the optimal treatment method for each individual case. We hope that this method will be developed in the future.

## Figures and Tables

**Figure 1 biomedicines-12-01411-f001:**
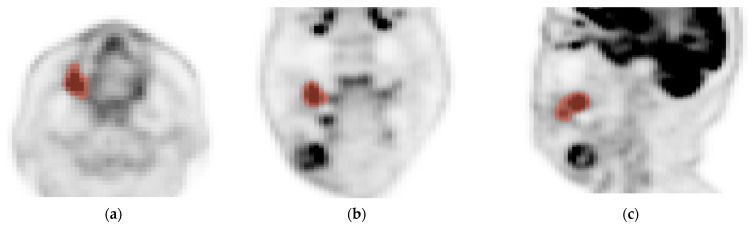
Example of segmentation on PET image of target patient. (**a**–**c**) are axial, coronal, and sagittal images, respectively. Brown area is segmented tumor area.

**Figure 2 biomedicines-12-01411-f002:**
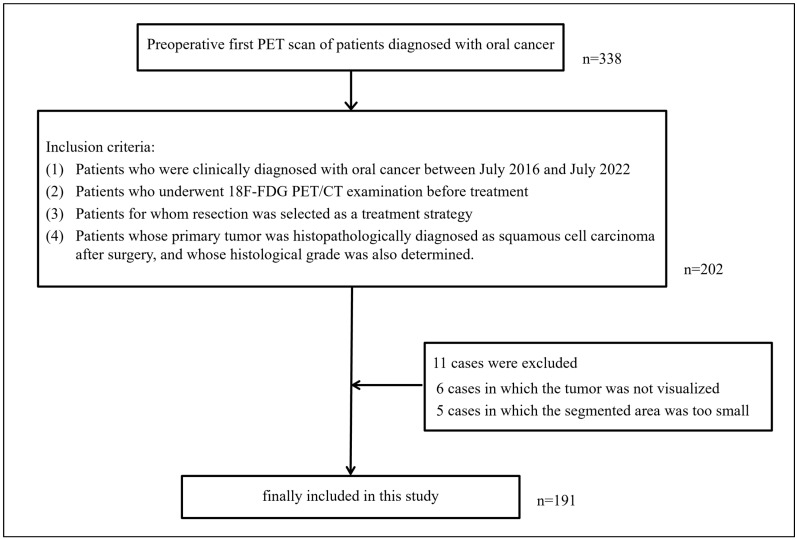
Flow chart of subject inclusion in this research.

**Figure 3 biomedicines-12-01411-f003:**
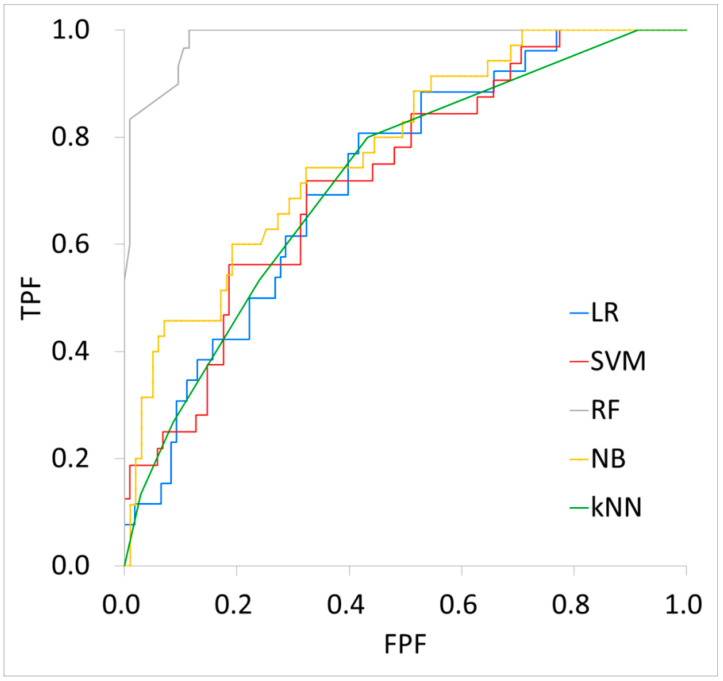
Performance of each machine learning model on training data. LR: Logistic Regression, SVM: Support Vector Machine, RF: Random Forest, NB: Bayes, kNN: k-Nearest Neighbor, TPF: True Positive Fraction, FPF: False Positive Fraction.

**Figure 4 biomedicines-12-01411-f004:**
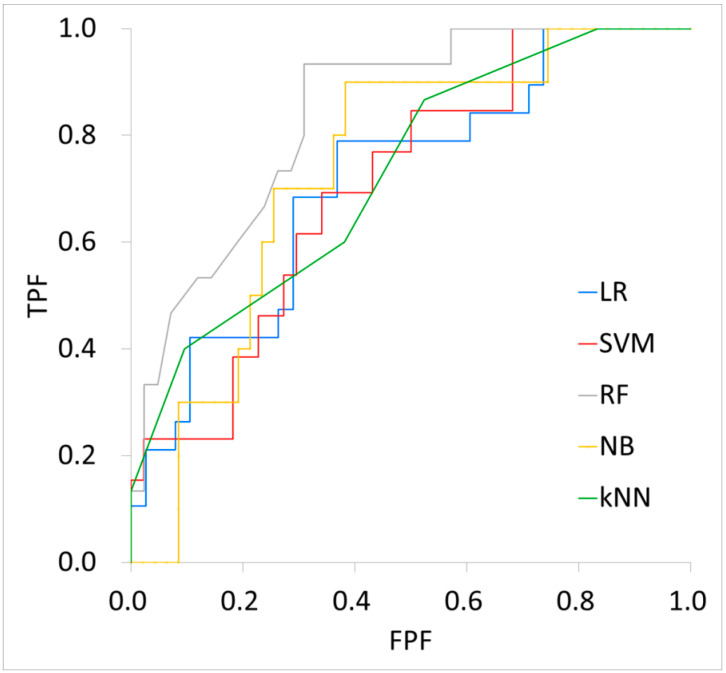
Performance of each machine learning model on testing data. LR: Logistic Regression, SVM: Support Vector Machine, RF: Random Forest, NB: Bayes, kNN: k-Nearest Neighbor, TPF: True Positive Fraction, FPF: False Positive Fraction.

**Table 1 biomedicines-12-01411-t001:** The characteristics of patients included in the present study.

Characteristics	Number
Age	
Average	68.9
Range	23–91
Sex	
Male	113
Female	78
Site of primary tumor	
Tongue	97
Floor of oral mouth	10
Gingiva of maxilla	22
Gingiva of mandible	38
Buccal mucosa	22
Palate	1
Lip	1
pT classification	
T1	69
T2	65
T3	12
T4a	42
T4b	3
Histological grade	
Well differentiated	146
Moderately/poorly differentiated	45

**Table 2 biomedicines-12-01411-t002:** Radiomics features with significant differences in Welch’s *t*-test.

Feature
bins0.1 original NGTDM Contrast
bins2 original GLSZM Zone Percentage
bins0.01 WL_HHH Histogram Mean
bins0.03 WL_HHH GLSZM Small Area High Gray Level Emphasis
bins0.05 WL_HHH GLDM Small Dependence High Gray Level Emphasis
bins0.05 WL_HHH GLSZM High Gray Level Zone Emphasis
bins0.05 WL_HHH GLSZM Small Area High Gray Level Emphasis
bins0.05 WL_HHH GLSZM Small Area Low Gray Level Emphasis
bins0.05 WL_HHH GLCM Cluster Shade
bins0.1 WL_HHH GLCM Cluster Shade
bins0.1 WL_HHH GLCM Difference Average
bins0.1 WL_HHH GLDM Small Dependence High Gray Level Emphasis
bins0.1 WL_HHH GLRLM Short Run High Gray Level Emphasis
bins0.1 WL_HHH GLRLM Gray Level Variance
bins0.2 WL_HHH GLCM Difference Average
bins0.2 WL_HHH GLCM Sum Squares
bins0.2 WL_HHH GLCM Contrast
bins0.2 WL_HHH GLCM Cluster Shade
bins0.2 WL_HHH GLCM Cluster Tendency
bins0.2 WL_HHH GLRLM Gray Level Variance
bins0.3 WL_HHH GLCM Contrast
bins0.3 WL_HHH GLCM Sum Squares
bins0.3 WL_HHH GLDM Gray Level Variance
bins0.3 WL_HHH GLRLM Gray Level Variance
bins0.3 WL_HHH GLSZM Gray Level Variance
bins0.5 WL_HHH GLCM Sum Average
bins0.5 WL_HHH GLCM Idmn
bins0.5 WL_HHH GLDM High Gray Level Emphasis
bins0.5 WL_HHH GLDM Low Gray Level Emphasis
bins0.5 WL_HHH GLRLM Short Run High Gray Level Emphasis
bins0.5 WL_HHH GLRLM Low Gray Level Run Emphasis
bins0.5 WL_HHH GLRLM High Gray Level Run Emphasis
bins0.5 WL_HHH NGTDM Complexity
bins5 WL_HHH GLRLM Short Run High Gray Level Emphasis
bins0.1 WL_LLL GLDM Dependence Non-Uniformity Normalized
bins5 WL_LLL GLSZM Zone Percentage
bins5 WL_LLL GLSZM Small Area High Gray Level Emphasis
bins10 WL_LLL GLSZM Small Area High Gray Level Emphasis
bins10 WL_LLL NGTDM Strength
Number of features
Bins
0.01–0.05 7 features
0.1–0.5 26 features
1–5 4 features
10 2 features
Image
Original image 2 features
Wavelet HHH image 32 features
Wavelet LLL image 5 features
Feature type
Shape feature 0 features
First order feature 1 features
Texture feature 38 features
Matrix of texture feature
GLCM 12 features
GLDM 6 features
GLRLM 8 features
GLSZM 9 features
NGTDM 3 features

**Table 3 biomedicines-12-01411-t003:** Selected features and their respective coefficients.

Feature	Coefficient
bins0.01 WL_HHH Histogram Mean	−0.00596191
bins0.05 WL_HHH GLSZM Small Area Low Gray Level Emphasis	−0.00429615
bins5 WL_HHH GLRLM Short Run High Gray Level Emphasis	−0.00345513
bins10 WL_LLL GLSZM Small Area High Gray Level Emphasis	−0.00285311
bins0.5 WL_HHH GLRLM Short Run High Gray Level Emphasis	−0.00199623
bins0.5 WL_HHH GLCM Idmn	−0.0019436
bins0.5 WL_HHH NGTDM Complexity	−0.00122783
bins0.05 WL_HHH GLCM Cluster Shade	−0.000958007
bins0.1 WL_LLL GLDM Dependence Non-Uniformity Normalized	0.0300688

**Table 4 biomedicines-12-01411-t004:** Performance of each machine learning model.

I Title 1	ML Model	AUC	Accuracy (%)	Sensitivity (%)	Specificity (%)	PPV (%)	NPV (%)
Training cohort (n = 134)	LR	0.72	81	12	98	60	82
	SVM	0.73	76	13	100	100	78
	RF	0.98	94	77	99	96	94
	NB	0.78	74	51	82	50	83
	kNN	0.72	78	0	100	0	78
Testing cohort (n = 57)	LR	0.72	67	11	100	100	100
	SVM	0.71	77	15	98	67	80
	RF	0.84	79	53	88	62	84
	NB	0.74	72	70	72	35	92
	kNN	0.73	74	13	100	100	76

ML model: machine learning model, LR: Logistic Regression, SVM: Support Vector Machine, RF: Random Forest, NB: Bayes, kNN: k-Nearest Neighbor, AUC: Area Under Curve, PPV: Positive Predictive, NPV; Negative Predictive Value.

## Data Availability

The data presented in this study are available on request from the corresponding author due to regulations in institutional research ethics review.

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
