# Peer review of "Prediction of Histological Grade of Oral Squamous Cell Carcinoma Using Machine Learning Models Applied to 18F-FDG-PET Radiomics"

_biomedicines, 2024, doi:10.3390/biomedicines12071411_

Round 1

Reviewer 1 Report

Comments and Suggestions for Authors

The manuscript by Nikkuni Y. et al. is well-written and interesting. They describe the histological grading of oral squamous cell carcinoma (OSCC) and compare it to a learning model using FDG-PET radiomics, aiming to avoid the need for biopsy in diagnosing the histological grade of OSCC. The authors evaluate different grades of OSCC-well, moderate, and poor, providing several clinical data in a table.

Despite the manuscript's interesting content and explanation of biopsy limitations, the study's purpose is not entirely clear. The authors attempt to compare a machine learning-based model with the evaluation of OSCC biopsy, but the objective lacks clarity. The methodology in point 2.2 is also unclear. Although the discussion presents interesting data, it immediately reveals that the model used is not fully functional compared to a biopsy diagnosis.

I suggest that the authors better describe their objectives, which would impact the title and discussion. They should also provide a more comprehensive conclusion highlighting the advantages of the studied model, such as aiding in diagnosis, predicting behavior, and associating histological diagnosis with clinical data. Additionally, it would be beneficial to review the WHO classification for 2022 (5th ed., 2022) for accuracy.

Reviewer 2 Report

Comments and Suggestions for Authors

Dear Authors,

The article entitled "Prediction of histological grade of oral squamous cell carcinoma using machine learning models applied to 18F-FDG-PET radiomics" comprises a well-done study on oral carcinoma and PET scanning. 

The article is original, comprehensive, good quality, and suitable for publication after a minor revision.

From the start, I want to mention that the tested hypothesis is not economically feasible(lines 73-74: "we hypothesized that radiomics features extracted from PET images can predict the histological grade of oral squamous cell carcinoma").  The costs of the histopathological evaluation are not comparable with PET scanning and the usage of machine learning algorithms to make predictions. Moreover, the oral cavity is easy to be examined for dentists and other medical specialists. This idea should be included as one main limitation of the present study.

You mentioned(line 396), "In the future, we would like to establish a method that shows higher accuracy in the differentiation of histological grades." which brings more light to the manuscript. You can add this idea in the Discussion chapter emphasizing the biomarkers and genetic research that can be used in this direction.

Best regards!

Round 2

Reviewer 1 Report

Comments and Suggestions for Authors

Nikkuni Y et al.'s second revision of the manuscript has improved over the first, and the authors have addressed all of my questions and suggestions. I recommend it without any changes.